# OpenReview forum: "Incentive-Aligned Multi-Source LLM Summaries"
_ICLR.cc/2026/Conference — ICLR 2026 Poster_

### Official Review · Reviewer_eWFW · 2025-10-29

**Soundness:** 3
**Presentation:** 4
**Contribution:** 3
**Rating:** 4
**Confidence:** 4

**Summary:**

The paper introduces Truthful Text Summarization (TTS), a multi-document text summarization framework that promotes peer-agreement among the sources on the facts in the summary and scores each source's stance on each atomic claim from the summary. In this way, TTS achieves summary faithfulness in noisy and adversarial conditions (where some documents are manipulative or noisy).

The authors provide a substantial mathematical framework of peer prediction applied to summarization under incentives.
They conduct experiments on two datasets, NaturalQuestions and ClashEval (300-document subsets) which they synthetically augment to contain faithful and unfaithful sources for each query. They use Gemini-2.5-flash as the LLM backbone and compare TTS to 3 baseline methods: Initial Summary (multi-doc summary), Majority Prompt (an LLM summary that only includes majority claims), and Majority Claims (majority claims are used to obtain a re-summary). In automatic evaluation, TTS strongly outperforms all the baselines in terms of Precision, Recall, F1 Score and Accuracy (calculated via an LLM-as-a-Judge).

**Strengths:**

* a novel technique is introduced that frames summarization with incentive as peer prediction
* a substantial theoretical framework is presented for informative corroboration among multiple sources
* the technique outperforms a series of baselines on NaturalQuestions and ClashEval synthetically adapted for the task

**Weaknesses:**

* evaluation datasets, although originally containing real-world data, heavily rely on data synthesis - both truthful sources (paraphrases) and untruthful ones are synthesized using an LLM. For evaluating a technique that claims to be robust against strategic misinformation / manipulation attacks, this does not seem to be enough
* evaluation would also be much more complete with human judgements which could expose what rather brittle P, R, F1 & LLMaaJ (a bit less so) cannot capture. ROUGE / BLEU for fluency evaluation are questionable

**Questions:**

N/A

---

> ### Author Response · Authors · 2025-11-19
> **Response to weakness 1 (synthetic data evaluation)**
>
> We thank the reviewer for the comments. We address each of the reviewer’s points in turn, and we thank the reviewer for engaging with our work.
>
> ----
> ### W1 (synthetic data evaluation):
>
> We understand the reviewer’s concern about synthetic data in the evaluation. Here we explain why we believe our current setup is a suitable and practical choice for the setting of this paper.
>
> Our use of synthetic sources is driven by the specific incentive environment we study. The paper targets a forward-looking regime with increasing adoption of LLM search systems (e.g., AI overviews), where content creators optimize for how an LLM summarizes multiple documents, not just for traditional link ranking. That ecosystem is only just emerging, so today’s web does not yet contain large amounts of systematically optimized “LLM-facing” content that we could simply reuse as a benchmark.
>
> To approximate this regime as realistically as we can, we:
> - Use real user queries and human-written long-form answers from NaturalQuestions and ClashEval; and
> - Generate truthful paraphrases, low-quality sources, and deceptive / prompt-injection sources with a strong deep-reasoning model (Gemini-2.5-Flash).
>
> This yields high-quality texts that explicitly implement the behaviors we care about—deceptive alternatives, prompt injections, and coordinated uninformative blocs, while remaining grounded in real queries and gold answers. We view this as a reasonable and somewhat conservative proxy for how sophisticated actors might attack AI overviews in the near future.
>
> Recognizing the reviewers’ points, we also tried to incorporate non-synthetic sources wherever possible.
>
> As discussed in the paper, we systematically examined verification / misinformation and RAG-style datasets (e.g., a partial list is given in the paper by Zeng et al., NeurIPS 2025, Table 1 https://arxiv.org/pdf/2502.16101) and looked specifically for data that (i) is query-driven, (ii) provides multiple conflicting contexts per query, and (iii) uses real web pages. The only dataset we found is QACC (Liu et al., 2024, ACL, https://arxiv.org/abs/2410.12311), which includes 1000+ examples where retrieved contexts may agree or conflict. After filtering for queries with at least 3 clearly truthful sources and at least 1 conflicting source, only 43 usable items remain. Moreover, when we re-checked the provided labels with a strong LLM (gemini-2.5-flash), about 23% of the labels were flagged as dubious, suggesting non-trivial human annotation noise.
>
> Despite these limitations, we still treat this dataset as a valuable practical example. To better match our strategic setting while keeping the non-synthetic nature of the pages, we leave each inaccurate document’s content intact and simply ask an LLM to append hidden prompt-injection instructions on top (mimicking real-world injection attacks in webpages). On this small but real-web subset, TTS still yields notable gains:
>
> **Open-Domain QA with Conflicting Contexts**
>
> | Method                       | Precision (%) | Answer Acc (%) |
> |-----------------------------|--------------|----------------|
> | **Baselines**               |              |                |
> | Baseline: Initial Synthesis | 64.7         | 46.3           |
> | Baseline: Majority Prompt   | 62.4         | 29.3           |
> | Baseline: Majority Claims   | 69.8         | 46.3           |
> | **Proposed Method**         |              |                |
> | **TTS**  | **80.6**         | **65.9**           |
>
> ----
>
> **Source Reliability Scores**
>
> | Source Name | Avg. Reliability (w_i) |
> |------------|-------------------------|
> | adversarial| 0.020806                |
> | deceptive  | 0.047381                |
> | truthful_1 | 0.064688                |
> | truthful_2 | 0.052313                |
> | truthful_3 | 0.048616                |
> | truthful_4 | 0.054240                |
>
> Hopefully, this practical example helps address the concern about synthetic data. It also illustrates why we rely on synthetic sources for the main evaluation:
>
> - Its “conflicts” most often arise from outdated facts or minor discrepancies (e.g., “January 25, 1994” vs. “1994”), rather than intentionally strategic or adversarial content.
> - The real dataset is small (43 items after filtering) and noisy (20%+ potential error on the human-annotated labels), making it unsuitable as a primary benchmark.
>
> By contrast, our synthetic setup lets us precisely instantiate the kinds of strategic, adversarial, and uninformative sources we care about, while still grounding everything in real queries and human gold answers. For a methodology paper whose main contribution is a new incentive-aligned mechanism, we believe this is an appropriate and informative evaluation choice. We explicitly frame the work as providing (i) a theoretical model and guarantees and (ii) empirical validation on a controlled setting, rather than as a new benchmark. We welcome future work that develops richer, real-world datasets for this emerging problem.

---

> ### Author Response · Authors · 2025-11-19
> **Response to weakness 2 (Human evaluation and metrics) Part 1**
>
> ----
>
> ### W2: On human evaluation and evaluation metrics, Part 1:
>
> We appreciate the reviewer’s concern about relying on automatic metrics using LLM-as-a-judge. At the same time, our experience in this project suggests that human labels themselves can be surprisingly brittle, and that a strong LLM-as-a-judge can, in our specific setting, be a reasonable—and sometimes cleaner and more accurate—alternative.
>
> Concretely, when we examined the Open-Domain QA with Conflicting Contexts dataset used in our small real-web experiment, we filtered it to queries with ≥3 clearly truthful sources and ≥1 conflicting source. On this filtered subset, a strong LLM (Gemini-2.5-Flash) flagged roughly 23% of the human-annotated labels as dubious on re-inspection. When we manually examined a sample of these, many (if not most) indeed appeared mislabeled. This is consistent with broader observations that human labels in open-domain QA and summarization, most of which sourced via crowdsourcing platforms like Amazon Mechanical Turk, can contain nontrivial noise, especially when instructions are subtle or when annotators must reconcile long, heterogeneous contexts. In other words, treating human labels as unquestionable ground truth is itself risky in this domain.
>
> By contrast, the main evaluation task for TTS is deliberately simple and closely aligned with what our mechanism optimizes. For each query, we have:
>
> - A short gold answer (a few tokens); and
> - A candidate summary produced by TTS or a baseline.
>
> The judge’s job is just: “Given the gold short answer and this summary paragraph, is the summary factually correct with respect to the gold answer?” Similarly, for the task of extracting precision and recall, we have the gold context as a paragraph and simply asks the language model to parse whether each claim is supported, and whether an original claim is featured in the summary. This is a fairly constrained semantic matching task, with clear objectives, inputs, and goals, much easier than free-form quality scoring, and we argue this is the type of tasks large language models are well-suited to evaluate.
>
> For this, we further use a state-of-the-art Deep Reasoning Model Gemini-2.5-Flash as an automatic judge with a full chain-of-thought budget. We believe that such a SOTA model, which performs well across the board and can even solve competitive math problems at top-human levels, will be able to perform the most natural semantic parsing problem that a LLM is expected to be good at fairly accurately. This design is supported by recent work on LLM-as-a-judge, which shows that strong LLM evaluators can reach substantially higher correlation with human judgments than surface-form metrics such as ROUGE/BLEU, and in some cases approach human–human agreement on factual and quality judgments for NLG tasks (e.g., Liu et al. 2023, https://arxiv.org/abs/2303.16634 with GPT-4).
>
> We are not claiming that LLM judges are perfect, only that for these binary correctness task on natural language domain—matching a summary to a short gold answer and examining if a claim is supported in a paragraph—they are at least as consistent as typical crowdworker annotations, and often less noisy, as they are exactly the type of tasks we expect a “Language Model” to do well on.
>
> As an additional sanity check, we (the authors) manually inspected a random sample of 100 NaturalQuestions examples along with the corresponding Gemini-as-judge “correct/incorrect” labels. In this spot-check, we did not find any disagreements between the judge’s decision and our own assessment of factual correctness. This is of course not a formal guarantee, but it gives us empirical confidence that, for the specific evaluation task we pose, the LLM judge is behaving sensibly.
>
> More broadly, our evaluation metrics are chosen to match what TTS is actually optimizing:
>
> - Claim-level precision/recall against human-written long-form answers, which directly measure whether the mechanism is selecting the right atomic facts; and
> - LLM-as-a-judge answer accuracy, which scales factual correctness checking to the full filtered NQ and ClashEval splits.
>
>
> We agree with the reviewer that human evaluation—especially targeted at nuanced phenomena like overly cautious abstention, subtle misleading phrasing, or user-perceived usefulness—would be a valuable complement, but we view it as a natural extension for future work and hope that we have persuaded the reviewer that the current metrics provides valid and insightful validation.

---

> ### Author Response · Authors · 2025-11-19
> **Response to weakness 2 (Human evaluation and metrics) Part 2**
>
> ----
>
> ### W2: On human evaluation and evaluation metrics, Part 2:
>
> As suggested, in the revision, we:
> - De-emphasize ROUGE/BLEU in the main text (keeping them as auxiliary fluency metrics in the appendix), and
> - Clarify that our primary quality signals are claim-level correctness and LLM-judged answer accuracy, with the LLM judge serving as a scalable, carefully sanity-checked proxy.
>
> Overall, we see our setup as complementary to traditional human studies: it trades some nuance for scale and reduced annotation noise. In a domain where existing human labels are themselves error-prone, a strong LLM-as-judge plus targeted manual spot-checks provides a reasonably reliable—and in some respects cleaner—signal for the mechanism-level questions we study.
>
> In case the reviewer is not totally persuaded, we provide the following illustrative example to further illustrate our point on why a LLM judge is suitable, reliable, and sufficient in our context and the metrics we report:
>
> From NaturalQuestions:
>
> > "query": Host of the late show who was once a correspondent for the daily show
>
> > "Short gold answer": "Stephen Colbert"
>
> > “baseline summary”: “The host of *The Late Show* who was once a correspondent for *The Daily Show* is Samantha Bee.\n\nSamantha Bee transitioned from her impactful role as a correspondent on *The Daily Show* to host a prominent late-night program. She joined *The Daily Show with Jon Stewart* on July 10, 2003, and quickly distinguished herself. Bee was not only the show's first non-U.S. citizen correspondent but also became its all-time longest-serving regular correspondent after surpassing Stephen Colbert in 2011…[long paragraph]…While other former *Daily Show* correspondents, such as Stephen Colbert, also went on to host late-night programs, including *The Late Show* on CBS, Samantha Bee is identified as the definitive answer to this specific query. Her journey from correspondent to a leading late-night host is a significant part of the program's legacy.”
>
> > LLM judge label: incorrect
>
> > “TTS summary”: Stephen Colbert is the host of *The Late Show* who was once a correspondent for *The Daily Show*.\n\nStephen Colbert had an impactful tenure as a correspondent on *The Daily Show*, serving from June 26, 1997, to September 26, 2005. During his nearly nine years on the program, he was known for segments such as \"Even Stephven\" and \"This Week In God.\" Colbert also hosted *The Daily Show* six times between 2001 and 2003. His time on the show made him the longest-serving correspondent until Samantha Bee surpassed his record in 2011 …[more details]…. Before assuming hosting duties for *The Late Show* on CBS, Colbert made his final appearance on *The Daily Show* on August 6, 2015, during Jon Stewart's farewell episode.
>
> > LLM judge label: correct
>
> This type of example illustrates both (i) why we believe the judge’s task is well-posed and tractable for a strong LLM (since the task is clear, and that the semantic nature of it falls in the comfort zone of LLM, let alone the state-of-the-art large model we used), and (ii) how the correctness signal it produces aligns with what our mechanism is designed to optimize.

---

> ### Author Response · Authors · 2025-11-19
> **Conclusion as well as other major changes**
>
> ----
> ### Other major changes
>
> We also like to provide a brief overview of some other major changes to the experimental section we made during the rebuttal to make our experiments more robust:
>
> To address other reviewers, we have scaled our primary evaluation to the **entire** Natural Questions (NQ-Dev) and ClashEval datasets, a four fold increase from original.
>
> - Expanded Dataset: We now evaluate on 1,131 datapoints for NQ and 1,299 datapoints for ClashEval.
> - Improved Methodology: We adopted the A/B grouping method (described in the earlier comment) for all new experiments. This not only standardizes the evaluation but significantly reduces computational complexity from $O(n^2m)$ to $O(nm)$ (where n is the number of sources and m is the number of claims), better aligning with real-world deployment requirements.
> - Results: The qualitative conclusions remain robust at scale. Across both datasets, our proposed method (TTS) substantially outperforms baselines in factual accuracy and claim-level precision/recall. Details are giving in Sec. 3 and 4 as well as App. L in the revised paper.
>
> ----
>
> We hope the reviewer find our response helpful, and we thank the reviewer for taking the time to read through this rebuttal. Please let us know if you have any remaining questions and we appreciate your feedback and further insights.

---

### Official Review · Reviewer_hbFR · 2025-10-31

**Soundness:** 2
**Presentation:** 2
**Contribution:** 2
**Rating:** 4
**Confidence:** 3

**Summary:**

This paper proposes TTS, a mechanism-design-based framework that incentivizes LLMs to prefer truthful over adversarial or uninformative sources during multi-source summarization. The core idea is to apply LOO peer-prediction scoring mechanism that assigns reliability scores to individual sources and rewards summaries consistent with high-reliability evidence. By reframing summarization as an incentive alignment problem rather than a post-hoc filtering task, the paper aims to structurally encourage honesty in LLM-generated summaries.

**Strengths:**

- Reframes summarization robustness as an incentive-alignment problem, introducing a peer-prediction mechanism rarely explored in LLM summarization.
- Demonstrates measurable suppression of uninformative and adversarial sources, yielding large factual-accuracy improvements.
- Provides empirical evidence that truthful reporting tends to emerge as the dominant strategy, as dishonest stances consistently reduce expected rewards.

**Weaknesses:**

- Although these assumptions (A1–A3) are standard in peer-prediction theory, they seem overly restrictive and arguably unnecessary for LLM summarization. Real-world sources are correlated through shared training data and retrieval biases, making the independence assumptions theoretically elegant but empirically weak.
- TTS requires each source to generate, cross-evaluate, and aggregate stances in a leave-one-out manner, leading to a quadratic number of model calls. The paper provides no runtime or scalability analysis, raising doubts about its practical feasibility for large-scale use.
- The evaluation relies mainly on internal ablations rather than strong external baselines. While TTS is positioned as conceptually distinct from retrieval-robust RAG systems, both aim to improve factual reliability under noisy evidence. Direct or qualitative comparisons with recent methods would better clarify TTS’s relative robustness and contribution.

**Questions:**

- What happens when all sources are unreliable? Does the system abstain or hallucinate?
- Does the incentive-based scoring scale to multi-hop or long-context summarization?
- Can TTS be integrated into the retrieval stage to detect adversarial documents earlier and promote a richer set of high-quality, reliable sources from the start?

---

> ### Author Response · Authors · 2025-11-19
> **Response to weakness 2 (computational complexity and scalability)**
>
> We thank the reviewer for the insightful comments. We address each of the reviewer’s points in turn, and we thank the reviewer for the time spent engaging with our work.
>
> ----
>
> ### W2 (Computational complexity and scalability):
>
> We appreciate the reviewer (and reviewer fpEF) for highlighting the computational cost of our original Leave-One-Out scoring scheme. This pushed us to revisit the design, and we incorporated a simple but important improvement:
>
> In the original paper, we utilized Leave-One-Out scoring: for each of the $n$ sources, we formed a held-out summary of its $n-1$ peers and decomposed that into claims for scoring. This guarantees that a source cannot shape the claims on which it is evaluated, which is key to the truthfulness guarantees, but leads to $O(n^2m)$ complexity, since we need to form $n$ different summaries for scoring each source, and for each summary we need to extract the stance of the $n$ sources on the $m$ claims.
>
> We realized that we can do much better. In the revision, we propose the following *A/B grouping scheme* that preserves the LOO spirit while reducing the complexity by a factor of $n$:
>
> - Randomly split the sources $\{1,2,...,n\}$ into two sets, $\mathcal{C}_A$ and $\mathcal{C}_B$.
> - For sources in $\mathcal{C}_A$, form the held-out summary and claim set from $\mathcal{C}_B$ only; for sources in $\mathcal{C}_B$, do the reverse.
>
> Using this method, each source is still evaluated on a claim set independent of its own document, but we only need to construct *two* total claim sets per query (one from $\mathcal{C}_A$, one from $\mathcal{C}_B$), *regardless of the number of sources*, giving overall complexity $O(nm)$ instead of $O(n^2m)$.
>
> We replace all of our main experiments with this new approach and empirically validate it on the datasets of Natural Questions and ClashEval (which we now evaluate on the **entire** dataset for both, with 1000+ samples each, for details see our comment below on W1). With 4 truthful sources, 1 low quality source, and 2 adversarial/deceptive sources, the results remain qualitatively unchanged from the original results and we’re able to demonstrate significant improvement in precision and accuracy using our TTS method.
>
> The main trade-off of the A/B scheme is that each claim set is formed from roughly $n/2$ sources instead of $n-1$. This can slightly reduce topical diversity when the number of sources is very small. In such cases, running full LOO is still the “first best”. However, as soon as there are a non-trivial number of sources, we find that the A/B scheme maintains qualitative performance while drastically improving scalability, and the empirical results remain solid.
>
> To fully understand the computational aspect of this new scheme, we provide a concrete runtime / cost estimate. For a typical query with 7 sources under the A/B scheme, the average token usage for each query for the mechanism is:
>
> **174,346** input tokens, **12,733** output tokens, **116,858** thinking tokens.
>
> Using current commercial cost for gemini-2.5-flash-lite, this is 0.017 dollars for input and 0.05 dollars for thinking+output each query, which we believe is a reasonable starting point at this time.
>
> We further note that
> - major search providers either host their own models or have discounted access, so real-world costs are likely much lower;
> - model efficiency and pricing trends are steadily improving. One of the major component of the cost above is the thinking tokens, and we’ll need much less of them to deliver the same performance when models improve.
>
> From a systems and practical perspective, there are additional levers:
>
> - All stance-extraction calls across claims and sources are totally parallelizable. In production, these would run concurrently, mitigating latency concerns; current AI-overview systems already tolerate several seconds of latency.
> - TTS does not need to be run on every single query: as we discuss in the paper (Appendix G), one can run TTS periodically (e.g., on popular queries or sampled traffic), aggregate per-source scores into a reputation signal, and then use this reputation to downweight low-scoring sources in retrieval and ranking. This slightly relaxes the per-query theoretical guarantees but is often acceptable in practice: once a source’s reputation deteriorates, the cost of maintaining deceptive content quickly outweighs the expected benefit.
>
> We have updated Section 3 and 4 to:
> - describe the A/B grouping scheme and its theoretical compatibility with our incentives analysis,
> - report the new full-dataset results with A/B scoring, and
> - include the concrete token and cost data above, along with a brief discussion of parallelization and periodic deployment.
>
> Overall, we believe this revision substantially strengthens the case that TTS is computationally feasible for realistic multi-source summarization settings.

---

> ### Author Response · Authors · 2025-11-19
> **Response to weakness 1 (Theoretical Assumptions)**
>
> ----
> ### W1 (Theoretical Assumptions):
>
> We adopt Assumptions A1–A3 because they match the canonical theoretical model in multi-task peer prediction and allow us to derive clear incentive and truthfulness guarantees in a simple, interpretable form. In other words, these assumptions are there to enable a rigorous analysis of the scoring rule; the practical pipeline does not mechanically require them in order to run.
>
> We agree with the reviewer that real-world sources are sometimes correlated. Our framework can address this in two ways:
>
> - **At the modeling level**:
> Assumption A2 is a conditional independence assumption: we assume that, once we fix the (latent) correctness of a claim and which sources actually speak about it, each source’s stance is drawn independently from its own noise channel. In other words, we explicitly allow sources to differ in accuracy, coverage, and bias, but we abstract away any additional dependence beyond that. This is the same abstraction used in standard multi-task peer-prediction models, and we use it to obtain a clean theorem.
>
> - **At the system level**:
> Practically, the strongest and most problematic correlations (near-duplicate or copy-pasted pages, multiple URLs with almost identical content, heavy domain-level repetition) are exactly the ones that can be more easily reduced by standard preprocessing such as semantic deduplication or per-domain caps. Our theoretical model should be read as describing this “deduplicated” regime: after obvious dependencies are filtered out, we analyze what happens if residual correlations are approximated as conditionally independent.
>
> Crucially, the pipeline itself does not require these assumptions to run; they are only used to obtain exact guarantees: it shows that under standard peer-prediction assumptions, our scoring rule makes truthful reporting the best strategy and punishes uninformative deviations. In practice, we do not expect every assumption to hold exactly, nor do we require the full strength of the guarantees to see benefit. What matters operationally is that the mechanism raises the cost and lowers the payoff of strategic behavior.
>
> Our experiments already operate in a correlated regime: truthful sources are all generated with the same base LLM and gold context, and thus share training data and stylistic artifacts. Even in this setting-where A1–A3 are at best approximate-TTS still cleanly separates reliable from unreliable sources and substantially improves factual accuracy. This suggests that, while the independence assumptions are an idealization for theory, the mechanism remains useful under the kinds of correlations that arise in realistic LLM-based summarization. We also added a more in depth discussion about the practicality of these assumptions in App. G.

---

> ### Author Response · Authors · 2025-11-19
> **Response to weakness 3 (Evaluation)**
>
> ----
>
> ### W3 (Evaluation):
>
> We thank the reviewer for raising this point. Our experimental design is centered on mechanism-level baselines because the main contribution of TTS is an incentive-aligned scoring rule for multi-source summarization, not a new end-to-end RAG architecture.
> Our three baselines are chosen to mirror the natural aggregation strategies available without access to ground truth or privileged priors:
>
> - Initial Synthesis ≈ a standard single-pass RAG-style summary over all sources.
>
>
> - Majority Prompt ≈ a prompt-level “consensus heuristic,” analogous to simple majority voting over evidence.
>
>
> - Majority Claims ≈ a claim-decomposition + majority-over-claims variant that enforces raw consensus without our peer-prediction scoring.
>
> These baselines let us isolate the effect of the peer-prediction–style scoring and filtering, i.e., what changes when we reward informative agreement rather than raw consensus, which is the core contribution.
>
> Most recent retrieval-robust RAG methods we survey in Appendix D (e.g., Self-RAG, Corrective RAG, AstuteRAG, MADAM-RAG / RAMDocs) are built around assumptions that are orthogonal to our focus:
> they either
>
> - rely on internal LLM priors, an LLM-as-judge module, or explicit labels to score / arbitrate content, or
>
> - focus on stronger prompting, debate, or self-critique at the generation layer while treating source incentives as fixed.
>
>
> In our target setting (open-web search with rapidly changing facts), we explicitly avoid assuming access to such trusted reference signals: static parametric knowledge can be stale, and fixed judges/prompts risk becoming “hidden rules” that strategic authors optimize against. If we simply plugged these systems in as direct baselines, we would have to either (a) give them exactly the kind of privileged signal we rule out in our model, or (b) run them without that signal, in which case they tend to collapse to behaviors very similar to the majority- or judge-style strategies we already include (Initial Synthesis / Majority-style baselines). In that sense, our current baselines are already capturing the non-oracle behavior of those pipelines in the environment we study.
>
> At the same time, we see these methods as complementary rather than competing. TTS is designed to be a modular source-filtering and reweighting layer: it can be inserted before or alongside retrieval-robust generation procedures to produce an incentive-aligned pool of documents, after which one could still apply Self-RAG, AstuteRAG, debate-style generation, etc. We have clarified this relationship in the revised related-work and experiments sections and explicitly flagged such combined systems as a promising direction for future work.
>
> ----
>
> We also wanted to highlight some other changes to the experimental section we made during the rebuttal to make our experiments more robust:
>
> To address other reviewers, we have scaled our primary evaluation to the **entire** Natural Questions (NQ-Dev) and ClashEval datasets, a four fold increase from original.
>
> - Expanded Dataset: We now evaluate on 1,131 datapoints for NQ and 1,299 datapoints for ClashEval.
> - Improved Methodology: We adopted the A/B grouping method (described in the earlier comment) for all new experiments. This not only standardizes the evaluation but significantly reduces computational complexity from $O(n^2m)$ to $O(nm)$, better aligning with real-world deployment requirements.
> - Results: The qualitative conclusions remain robust at scale. Across both datasets, our proposed method (TTS) substantially outperforms baselines in factual accuracy and claim-level precision/recall. Details are giving in Sec. 3 and 4 as well as App. L in the revised paper.
>
> We also evaluate on real examples of conflict in web sources, from the dataset Open Domain Question Answering with Conflicting Contexts (Liu et al., 2024, ACL, https://arxiv.org/abs/2410.12311). TTS still yields notable gains in factual accuracy and effectively downweights the adversarial source. Details are given in App. L.

---

> ### Author Response · Authors · 2025-11-19
> **Response to question 1 (all sources unreliable), q2 (multi-hop, long context) and q3 (retrieval integration)**
>
> ----
>
> ### Q1 (All sources unreliable):
>
> If all retrieved sources are unreliable and we deliberately do not use any external priors or ground-truth labels, then no purely peer-based method can reliably recover the truth - ours included. In our formalism this is exactly the regime ruled out by Assumption A3, which requires at least a small amount of informative signal on average in the source pool. When A3 fails (all sources uninformative or systematically wrong), the system will still produce an answer, but that answer is not anchored by any trustworthy evidence; this is a fundamental limitation of the information setting rather than a specific failure mode of TTS.
>
> We do not view this as a drawback of the mechanism so much as a reflection of the scope of the paper:
> - Our goal here is to study what can be done given some informative signal in the sources, without assuming access to a trusted oracle or to static LLM priors that may be stale in fast-moving domains.
> - The TTS pipeline is fully compatible with additional verification or prior-based components. In practice, TTS outputs a per-source score and filters out those below a threshold; any verification routine or prior-based method from related work can be applied before or after this filtering step, or treat the LLM’s own prior as just another scored “source.”
>
> To summarize: our theory requires at least some informative sources (A3), which is unavoidable in any label-free setting, and in practice TTS is designed as a composable building block rather than a complete replacement for all other safeguards. This is related to our response to q1 to reviewer FpEF, which might further clarify.
>
> ----
> ### Q2: Multi-hop and long-context:
>
> Conceptually, our framework is agnostic to whether the underlying summarization task is single-hop, multi-hop, or long-context. The scoring mechanism only interacts with:
> - the set of atomic claims produced by the decomposer $D$, and
>
> - the stance matrix $r_{ik}$ extracted by $E$ for each source–claim pair.
>
>
> In a multi-hop or long-context setting, the draft answer $M(\mathcal{C})$ will simply contain more complex reasoning chains, and D will produce a richer set of claims (potentially with larger K). The scoring rule and the theoretical guarantees do not need to change; if anything, having more claims is beneficial for us, because the concentration bounds in our analysis improve as K grows.
>
> The main limitation is computational, not conceptual: longer contexts and multi-hop reasoning will generally increase the number of claims and stance queries per source. Earlier, we already propose and validate an A/B-group variant of our LOO construction. In practice, further engineering tricks—such as hierarchical claim decomposition (score high-level claims first and only expand contentious ones), or sampling a subset of claims for scoring—can further control cost in this case.
>
> We do not yet evaluate on dedicated multi-hop / long-context benchmarks, and we explicitly flag this in App G.1 as an important direction for future work. But the incentive mechanism itself applies unchanged as long as we can (i) decompose answers into atomic claims and (ii) extract per-source stances for those claims.
>
> ----
>
> ### Q3 (Retrieval integration):
>
> Yes—this is a natural extension of our framework, and it fits well with the “reputation” (App. G) view we discuss for scalability.
>
> In the current paper, TTS is applied post-retrieval, per query: given a retrieved set $\mathcal{C}$, we assign per-source scores and use them to filter and reweight sources for that query’s summary. However, these per-query scores can be straightforwardly aggregated over time into a cross-query reputation signal:
> - Every time TTS runs, it produces a reliability score for each source (URL, domain, or source cluster).
> - These scores can be accumulated into a reputation prior as mentioned above and in App. G.1.
> - The retriever can then use this prior as an additional feature to downweight consistently low-scoring or adversarial sources and upweight sources that repeatedly score well.
>
>
> In this way, TTS naturally feeds into a retrieval-stage defense: adversarial or low-information documents identified by our mechanism will tend to be retrieved less often over time, and high-quality sources will be surfaced more frequently. As mentioned earlier, this also further alleviates the problem of computational costs.
>
> From a theory standpoint, integrating TTS into retrieval introduces temporal dynamics (scores today changing exposure tomorrow), which goes beyond the static per-query model we analyze in this paper; formally modeling that repeated-game setting is interesting future work.
>
> From a systems standpoint, however, there is no obstacle: TTS already outputs the source-level scores needed to plug into retrieval, and can serve as an upstream filter and reputation module for standard retrievers.
>
> We added the discussion above in App. G.1 to make this retrieval-integration point explicit

---

> > ### Author Response · Authors · 2025-11-19
> >
> > We hope the reviewer find our response helpful, and we thank the reviewer for taking the time to read through this rebuttal. Please let us know if you have any remaining questions and we appreciate your feedback and further insights.

---

> > > ### Comment · Reviewer_hbFR · 2025-11-26
> > >
> > > Thank you for the detailed response and thorough revision work. The improvement in computation costs(W2) from $O(n^2m)$ to $O(nm)$ is quite impressive with comparable performance to the prior method. I also agree that the robustness seems to come from unreliable sources being the minority. I still have concern about W3, but your explanation about the methodological differences and how your design can complement RAG methods helped me understand the concerns much better.
> > > The expanded experiments with larger datasets and an additional real-world dataset experiment in appendix, make the empirical validation much stronger. Since you addressed my concerns thoroughly, i revised my score.

---

> > > > ### Author Response · Authors · 2025-11-26
> > > > **Thank you for your response**
> > > >
> > > > We sincerely thank the reviewer for the positive feedback on our revision, and for highlighting the improvements in computational complexity and experimental scale. We are glad that the revised approach and the expanded experiments addressed your concerns, and we deeply appreciate you taking the time to re-evaluate our work and revise your score.

---

### Official Review · Reviewer_fpEF · 2025-10-31

**Soundness:** 3
**Presentation:** 3
**Contribution:** 3
**Rating:** 6
**Confidence:** 3

**Summary:**

This paper explores the important challenge of building LLM-based RAG summarization systems that are incentive-robust. They propose TTS, a mechanism for decomposing summaries into claims and using hold-one-out evaluate of each source to determine whether it scores highly enough under a peer-consistency-based reliability score to remain in the retrieved context.
The authors build a sophisticated theoretical framework modeling source claims, player strategic incentives, and the mechanisms for whether a claim is reported in order to show that the approach incentivizes truth-telling from all sources.
The approach is evaluated over 300 QA pairs from two datasets (NQ and ClashEval) for which the authors construct synthetic scenarios in which sources have mixed truthfulness.

**Strengths:**

S1. The proposed algorithm addresses an important, understudied problem of incentive alignment and is of interested to researchers working on creating adversary-robust RAG approaches.

S2. The authors provide substantial valuable theoretical backing to show that the proposed approach disincentivizes false reporting. The extent of the analysis is comendable.

S3. Experiments model multiple interesting scenarios, including coordinated and uncoordinated adversarial behavior. In these scenarios, the results seem convincing.

**Weaknesses:**

W1. The experiments include only 300 datapoints for 2 datasets; this is substantially less than a typical robust evaluation. The paper could benefit from larger datasets and at least one more, noticeably different source/domain for which RAG models are typically used. The scenarios constructed from these datasets are also synthetically generated and not necessarily reflective of real world RAG QA. The Sources (reliable, deceptive, and adversarial) are all LLM generated and may have artifacts that are easier to detect (i.e. less subtle) than real ones.

W2. The approach is substantially slower and more computationally intensive than typical RAG -- it requires generating stance classifications for every (claims, document) in the cross-product of claims/documents in all retrieved sources except a claims's original source. (The claims also need to be extracted, but this can be pre-computed in a RAG index). The paper could benefit from a real-world case study.

W3. The writing can often be dense and inaccessible. While the framework is interesting, it reads as aggressively over-mathematized. The authors could benefit from providing simple, prosaic, direct summaries of components of the theoretical framework before diving into some of the equations, some of which might be better appendixed. Some of the wording is also generally difficult to digest; the paragraph that conveys some of the contributions P80-86, is jargon-heavy and quite difficult to understand even after multiple passes by a native speaker with a PhD in the field. e.g. these lines are hard to read: L.83-85 "Signals are embedded in prose: [...] formalize implementability and an equivalence to the standard signal–report model."

**Questions:**

Q1. Does source selection that disincentivizes a source from having a unique claim end up hurting downstream QA because it decreases diversity in the retrieved context?

Q2. The theoretical analysis makes the modeling assumption that all claims are exchangeable, but is this ever true in practice? Some kinds of sources are likely avoid claims about controversial/uncertain topics and are more likely to report claims that are newsworthy. Therefore it seems that $Pr(Q_{ik} = 1 \mid T_i)$ cannot be boiled down to a constant, it is dependent on a claim's type (e.g. "controversial", "speculative", etc) that is not exchangeable.

---

> ### Author Response · Authors · 2025-11-19
> **Response to weakness 2 (Computational complexity)**
>
> We thank the reviewer for the insightful comments and positive feedback. We address each of the reviewer’s points in turn, and we thank the reviewer for the time spent engaging with our work.
>
> ***
>
> ### W2 (Computational complexity and scalability)
>
> We appreciate the reviewer (and reviewer hbFR) for highlighting the computational cost of our original Leave-One-Out scoring scheme. This pushed us to revisit the design, and we incorporated a *simple but important improvement*:
>
> In the original paper, we utilized Leave-One-Out scoring: for each of the $n$ sources, we formed a held-out summary of its $n-1$ peers and decomposed that into claims for scoring. This guarantees that a source cannot shape the claims on which it is evaluated, which is key to the truthfulness guarantees, but leads to $O(n^2m)$ complexity, since we need to form $n$ different summaries for scoring each source, and for each summary we need to extract the stance of the $n$ sources on the $m$ claims.
>
> We realized that we can do much better. In the revision, we propose the following *A/B grouping scheme* that preserves the LOO spirit while reducing the complexity by a factor of $n$:
>
> - Randomly split the sources $\{1,2,...,n\}$ into two sets, $\mathcal{C}_A$ and $\mathcal{C}_B$.
> - For sources in $\mathcal{C}_A$, form the held-out summary and claim set from $\mathcal{C}_B$ only; for sources in $\mathcal{C}_B$, do the reverse.
>
> Using this method, each source is still evaluated on a claim set independent of its own document, but we only need to construct *two* total claim sets per query (one from $\mathcal{C}_A$, one from $\mathcal{C}_B$), *regardless of the number of sources*, giving overall complexity $O(nm)$ instead of $O(n^2m)$.
>
> We replace all of our main experiments with this new approach and empirically validate it on the datasets of Natural Questions and ClashEval (which we now evaluate on the **entire** dataset for both, with 1000+ samples each, for details see our comment below on W1). With 4 truthful sources, 1 low quality source, and 2 adversarial/deceptive sources, the results remain qualitatively unchanged from the original results and we’re able to demonstrate significant improvement in precision and accuracy using our TTS method.
>
> The main trade-off of the A/B scheme is that each claim set is formed from roughly $n/2$ sources instead of $n-1$. This can slightly reduce topical diversity when the number of sources is very small. In such cases, running full LOO is still the “first best”. However, as soon as there are a non-trivial number of sources, we find that the A/B scheme maintains qualitative performance while drastically improving scalability, and the empirical results remain solid.
>
> To fully understand the computational aspect of this new scheme, we provide a concrete runtime / cost estimate. For a typical query with 7 sources under the A/B scheme, the average token usage for each query for the mechanism is:
>
> **174,346** input tokens, **12,733** output tokens, **116,858** thinking tokens.
>
> Using current commercial cost for gemini-2.5-flash-lite, this is 0.017 dollars for input and 0.05 dollars for thinking+output each query, which we consider to be a reasonable starting point at this time.
>
> We further note that
> - major search providers either host their own models or have discounted access, so real-world costs are likely much lower;
> - model efficiency and pricing trends are steadily improving. One of the major component of the cost above is the thinking tokens, and we’ll need much less of them to deliver the same performance when models improve.
>
> From a systems and practical perspective, there are additional levers:
>
> - All stance-extraction calls across claims and sources are totally parallelizable. In production, these would run concurrently, mitigating latency concerns; current AI-overview systems already tolerate several seconds of latency.
> - TTS does not need to be run on every single query: as we discuss in the paper (Appendix G), one can run TTS periodically (e.g., on popular queries or sampled traffic), aggregate per-source scores into a reputation signal, and then use this reputation to downweight low-scoring sources in retrieval and ranking. This slightly relaxes the per-query theoretical guarantees but is often acceptable in practice: once a source’s reputation deteriorates, the cost of maintaining deceptive content quickly outweighs the expected benefit.
>
> We have updated Section 3 and 4 to:
> - describe the A/B grouping scheme and its theoretical compatibility with our incentives analysis,
> - report the new full-dataset results with A/B scoring, and
> - include the concrete token and cost data above, along with a brief discussion of parallelization and periodic deployment.
>
> Overall, we believe this revision substantially strengthens the case that TTS is computationally feasible for realistic multi-source summarization settings.

---

> ### Author Response · Authors · 2025-11-19
> **Response to weakness 1 (Part 1)**
>
> ----
> ### W1 (Experiment scale and synthetic sources) Part 1
>
> We thank the reviewer for raising concerns about evaluation scale and the use of synthetic sources. We address both points here.
>
> **Scale**
>
> The reviewer noted that 300 datapoints may be insufficient for a robust evaluation. To address this, we have scaled our primary evaluation to the **entire** Natural Questions (NQ-Dev) and ClashEval datasets [1].
>
> - Expanded Dataset: We now evaluate on 1,131 datapoints for NQ and 1,299 datapoints for ClashEval.
> - Improved Methodology: We adopted the A/B grouping method (described in the earlier comment) for all new experiments. This not only standardizes the evaluation but significantly reduces computational complexity from $O(n^2m)$ to $O(nm)$, better aligning with real-world deployment requirements.
> - Results: The qualitative conclusions remain robust at scale. Across both datasets, our proposed method (TTS) substantially outperforms baselines in factual accuracy and claim-level precision/recall. We updated the results in the new paper and also provide them below for convenience.
>
> ----
>
> **Table 1: Natural Questions (Experiment 1)**
>
> | Method                       | Precision (%) | Recall (%) | F1-Score (%) | Answer Acc (%) |
> |-----------------------------|--------------|------------|--------------|----------------|
> | **Baselines**               |              |            |              |                |
> | Initial Synthesis           | 40.8         | 26.2       | 31.9         | 25.1           |
> | Majority Prompt             | 43.4         | 26.5       | 32.9         | 27.5           |
> | Majority Claims             | 50.1         | 24.7       | 33.1         | 38.6           |
> | **Proposed Method**         |              |            |              |                |
> | **TTS (A/B Group Filtering)**  | **76.1**         | **37.2**       | **50.0**        | **72.3**           |
>
>
> ----
>
>
> **Table 2: Natural Questions (Experiment 1) - reliability scores**
>
> | Source Name | Avg. Reliability (w_i) |
> |------------|-------------------------|
> | truthful_2 | 0.1079                  |
> | truthful_1 | 0.1072                  |
> | truthful_3 | 0.1068                  |
> | truthful_4 | 0.1033                  |
> | low_quality    | 0.0394                  |
> | adversarial| 0.0325                  |
> | deceptive  | 0.0160                  |
>
>
> ----
>
> **Table 3: ClashEval (Experiment 1)**
>
> | Method                       | Precision (%) | Recall (%) | F1-Score (%) | Answer Acc (%) |
> |-----------------------------|--------------|------------|--------------|----------------|
> | **Baselines**               |              |            |              |                |
> | Initial Synthesis           | 49.3         | 22.0       | 30.4         | 15.6           |
> | Majority Prompt             | 58.7         | 25.4       | 35.4         | 30.2           |
> | Majority Claims             | 63.6         | 20.4       | 30.9         | 38.4           |
> | **Proposed Method**         |              |            |              |                |
> | **TTS (A/B Group Filtering)**  | **86.2**         | **31.5**       | **46.1**         | **77.1**           |
>
> ----
>
> **Table 4: ClashEval (Experiment 1) - reliability scores**
>
> | Source Name | Avg. Reliability (w_i) |
> |------------|-------------------------|
> | truthful_1 | 0.0961                  |
> | truthful_3 | 0.0957                  |
> | truthful_2 | 0.0938                  |
> | truthful_4 | 0.0951                  |
> | low_quality    | 0.0526                  |
> | adversarial| 0.0207                  |
> | deceptive  | 0.0162                  |
>
>
> ----
>
> **Table 5: Experiment 2: Robustness against coordinated, uninformative behavior**
>
> | Method                      | Precision (%) | Recall (%) | F1-Score (%) | Answer Acc (%) |
> |----------------------------|--------------|------------|--------------|----------------|
> | Baseline (All Sources)     | 56.9         | 22.6       | 32.4         | 7.7            |
> | Majority Vote (LOO Filter) | 34.6         | 9.6        | 15.1         | 9.5            |
> | **Proposed Method**         |              |            |              |                |
> | **PeerPred (A/B group, TTS)** | **87.8**         | **28.8**       | **43.4**         | **74.2**           |
>
> ----
>
> **Table 6: Experiment 2: reliability scores**
>
> | Source Name | PeerPred (TTS) w_i | Majority Vote w_i |
> |------------|--------------------|-------------------|
> | truthful_1 | 0.0308             | -0.2777           |
> | truthful_2 | 0.0302             | -0.2665           |
> | truthful_3 | 0.0315             | -0.2066           |
> | adversarial| 0.0022             | 0.6289            |
> | uninformative1     | 0.0005             | 0.9292            |
> | uninformative2     | 0.0004             | 0.9310            |
> | uninformative3     | 0.0006             | 0.9433            |
> | uninformative4     | 0.0003             | 0.9404            |
>
> [1]: subject to the standard preprocessing as described in App. L of the paper.

---

> ### Author Response · Authors · 2025-11-19
> **Response to weakness 1 (Part 2)**
>
> ----
> ### W1 (Experiment scale and synthetic sources) Part 2
>
> **Why synthetic strategic sources**
>
> We appreciate the reviewer’s emphasis on realistic data. However, we argue that synthetic sources are necessary to capture the strategic incentives that are central to our mechanism design. As the reviewer notes, our paper takes a different, understudied angle:  we study how changing incentives in the emerging AI-overview / RAG+LLM search paradigm affects source behavior. Our goal is not only to make summaries robust given a fixed pool of documents, but to design a mechanism that makes it unprofitable for sources to behave strategically in the first place, in a world where AI overview becomes a dominant search paradigm.
>
> Today’s web has not yet fully adapted to that incentive environment. AI Overviews are in their exploratory, nascent stages; and traditional search still dominates, and we do not yet see large-scale, systematically optimized “LLM-facing” documents at the level we anticipate if AI overviews become the default entry point. As a result, there is currently no existing “fully realistic” dataset that exactly matches our setting. However, with the rapid adoption of these AI Overview systems and the increasing user conversion, we believe that we are not far from such a time.
>
> To approximate this future regime, we therefore construct strategic sources in our main experiments by:
> - Using real user queries and human-written long-form gold answers (NQ, ClashEval); and
> - Generating truthful paraphrases, low-quality sources, and deceptive / prompt-injection sources with a state-of-the-art deep reasoning model (Gemini-2.5-Flash), which is capable of writing content at or above typical human quality.
>
> We believe this is a reasonable (and in some sense conservative) proxy for how sophisticated content authors might behave when optimizing for AI-overview visibility in the near future.
>
> Having said the above, we recognize the reviewer’s point on the practicality of sources, and we tried our best to address this issue. As summarized in Appendix D, we systematically examined existing verification / misinformation and RAG datasets (e.g., a partial list is given in the paper by Zeng et al., NeurIPS 2025, Table 1 https://arxiv.org/pdf/2502.16101) and looked specifically for datasets that: (1) are grounded in user queries; (2) Provide multiple pieces of conflicting evidence per query; (3) are sourced from ‘real’ sources online.
>
> The only dataset we found that is even partially aligned is Open Domain Question Answering with Conflicting Contexts (Liu et al., 2024, ACL, https://arxiv.org/abs/2410.12311), which includes 1000+ real examples where retrieved contexts may agree or conflict. After filtering for queries with at least 3 clearly truthful sources and at least 1 conflicting source, we are left with only 43 usable items. Moreover, when we used a strong LLM (gemini-2.5-flash) to re-check the provided labels, about 23% of the labels were flagged as dubious (when we check manually, many, if not all, of those labeled dubious by the LLM is indeed inaccurate), suggesting non-trivial human annotation noise.
>
> Despite these limitations, we still treat this dataset as a valuable practical example. We added a new experiment on this dataset (details given in App. L5)  and found that on this small but real-web subset, TTS still yields notable gains in factual accuracy and effectively downweights the adversarial source:
>
> | Method                       | Precision (%) | Answer Acc (%) |
> |-----------------------------|--------------|----------------|
> | **Baselines**               |              |                |
> | Baseline: Initial Synthesis | 64.7         | 46.3           |
> | Baseline: Majority Prompt   | 62.4         | 29.3           |
> | Baseline: Majority Claims   | 69.8         | 46.3           |
> | **Proposed Method**         |              |                |
> | **TTS**  | **80.6**         | **65.9**           |
>
> Hopefully, this practical example helps address the concern about synthetic data. It also illustrates why we rely on synthetic sources for the main evaluation: Its “conflicts” most often arise from outdated facts or minor discrepancies (e.g., “January 25, 1994” vs. “1994”, or ambiguous answers), rather than intentionally strategic, deceptive, or adversarial content.
> The real dataset is small (43 items after filtering) and noisy (20%+ potential error on the human-annotated labels), making it unsuitable as a primary benchmark.
>
> Our synthetic setup, by contrast, lets us precisely instantiate the kinds of strategic, adversarial, and uninformative sources we care about, while still grounding everything in real queries and human gold answers. For a methodology paper whose main contribution is a new incentive-aligned mechanism, we believe this is an appropriate and informative evaluation choice. We welcome, and hope to motivate, future work that develops richer, real-world datasets for this emerging problem.

---

> ### Author Response · Authors · 2025-11-19
> **Response to weakness 3 (writing) and question 2 (theoretical assumptions)**
>
> ----
> ### W3 (Dense writing and math) :
>
> We appreciate the reviewer’s suggestion and agree that a clearer high-level presentation improves readability. In the revision we made the following concrete changes:
>
> - New high-level overview (Section 2.1 and 2.2). Before introducing the formal notation, we now give a high-level description of the generative model and mechanism, and the different components of the model. This section explains, in plain language, what the “signals,” “reports,” “strategy” are and how they relate to the QA setting, before any equations appear.
>
> - Rewriting the contribution paragraph (old L80–86, new L83-91). We have rewritten this paragraph to avoid jargon such as the ones the reviewer pointed out.
>
> - De-densifying the main text. We tried to lightly make the math more accessible and moved some of the heavier math to the appendix.
> Due to space and time constraints we focused our edits on the locations the reviewer identified as particularly hard to read (including L80–86, beginning of Section 2) and nearby expository bottlenecks. We will continue to polish the wording and organization for the camera-ready version, and we welcome further suggestions on specific places where additional simplifications would be helpful.
>
> ***
>
> ### Q2 (Theoretical Assumptions).
>
> We agree with the reviewers point that the homogeneity and conditional independence are strong. We adopt them for the same reason as most work in multi-source peer prediction: they give us a clean, tractable model in which we can prove an exact incentive-compatibility guarantee.
> In our analysis, these assumptions are used only at the point where we connect the peer-consistency score to latent source accuracy and show that truthful reporting is the source’s best response. Homogeneity/exchangeability lets us average over claims and get concentration; conditional independence lets us interpret agreement/disagreement between sources as information about their reliability rather than about shared noise. This yields a rigorous theorem and guarantees, but we do not claim that the web literally satisfies this model.
> In practice, our bar is lower and our evidence is empirical: we do not need to perfectly eliminate all strategic behavior, only to make systematic lying or strategic behavior much more costly and therefore much rarer. Our experiments already operate in a regime where the assumptions are at best approximate (LLM-generated sources, realistic QA queries), and we still observe that TTS substantially reduces the impact of deceptive/adversarial sources. We added this as a discussion in App. G.
>
> Regarding the reviewer’s point about different “types” of claims (e.g., controversial vs. routine facts): in a deployed system, one practical adaptation might be to first assign each claim to a small number of coarse categories (e.g., using a classifier or LLM), run our scoring mechanism separately within each category, and then combine the resulting scores (e.g., by averaging). Under the same conditions as in our theory, truth-telling maximizes the expected score within each category (if one assume the classifier is accurate). Because the total score is just a (weighted) sum of category scores, the strategy that maximizes each category-wise score is also the strategy that maximizes the overall score - namely, being truthful. This clustered variant is beyond the scope of the present analysis, but we see it as a practical extension.

---

> ### Author Response · Authors · 2025-11-19
> **Response to question 1 (uniqueness and diversity)**
>
> ***
>
> ### Q1 (Uniqueness and Diversity):
>
> We first want to emphasize that our mechanism does not penalize uniqueness per se. The stance model distinguishes between contradiction and mere lack of topic overlap: if a source contributes a claim that is not mentioned elsewhere but does not conflict with the shared core of other sources, it can be labeled as “NO_STANCE” rather than “CONTRADICT.” Such unique-but-compatible claims are still eligible to be included and, in practice, are synthesized into the final answer if the source passes the threshold. So a source is not punished just for saying something extra that others do not say.
>
> The more delicate case is when a unique claim directly contradicts a confident majority: e.g., ten sources assert A and one source asserts B, with similar apparent quality and A & B mutually exclusive. In our framework we explicitly assume that we do not have access to ground truth and we do not rely on static priors (e.g., asking the same LLM to arbitrate using its internal world knowledge), which we believe mirrors the regime of rapidly evolving search. In this model, we deliberately restrict ourselves to using only the mutual information between sources: the scoring rule, motivated by multi-task peer prediction, rewards sources that are mutually informative and uses the off-task term to discount uninformative “herd” equilibria. But if all the observable evidence is “10 say A, 1 says B” with no additional prior signal, there is no purely interaction-based reason to systematically favor B over A.
>
> Our goal in this paper is to analyze that interaction-based mechanism in isolation, not to design a full end-to-end retrieval stack. In practical deployments, it is natural to combine our mechanism with simple prior-based components when desired, and our design is explicitly compatible with that. For example:
>
> -  Reputation weighting (App. G): we already discuss a reputation-weighted variant where practitioners can assign higher prior weight to known reliable outlets (e.g., major newspapers) than to low-credibility sources (e.g., anonymous blogs). This provides a principled and explicit way to inject domain knowledge, so that a single high-reputation dissenting source can count more than low-reputation ones, if desired.
>
>
> - Composability with other routines: Our method is modular and can be applied on top of existing retrieval and filtering steps (e.g., domain filters, de-duplication, or diversity-promoting selection). In that case, TTS is used to score and prune among sources that have already passed those prior-based filters, rather than acting as the sole decision mechanism.
>
>
> We now added a subsection on this discussion in the paper (App. G).
>
> ----
>
> We hope the reviewer find our response helpful, and we thank the reviewer for taking the time to read through this rebuttal. Please let us know if you have any remaining questions and we appreciate your feedback and further insights.

---

> ### Comment · Reviewer_fpEF · 2025-11-26
>
> Thank you for addressing my concerns. I have updated my score accordingly.

---

> > ### Author Response · Authors · 2025-11-26
> > **Thank you for your response**
> >
> > We are delighted to hear that our revisions have addressed your concerns. We sincerely appreciate your thoughtful engagement and thank you for your positive assessment of our work.

---

### Author Response · Authors · 2025-11-30
**Executive summary of the rebuttal discussions (Pre-Rollback scores 8/6/4) and reviewer feedback**

Dear Area Chair,

We understand you have inherited this paper under extraordinary circumstances. We sincerely thank you for your service and for taking on these additional responsibilities during this challenging time.

We are writing to provide an executive summary of our rebuttal outcomes. Due to the unfortunate set of events leading to all scores being reset to pre-rebuttal, the current pre-rebuttal scores (6/4/4) *do not* reflect the positive revaluation and majority support reached prior to the system reset.

----

### Updates before system reset

Before the discussion period was cut short and scores reverted to pre-rebuttal, **two** reviewers explicitly raised their scores by 2 points, **resulting in an updated score of 8, 6, 4.** (**The third reviewer, initially a 4, had not yet responded to our comprehensive rebuttal**). We respectfully point you to the reviewers' comments, which confirms the following updates on November 26th *(an entire day prior to the reports of the system bug on Nov 27th)*:

Reviewer fpEF (Score: 6 →  8):

> Quote (Nov 26): "Thank you for addressing my concerns. I have updated my score accordingly."

Reviewer hbFR (Score: 4 →  6):

> Quote (Nov 26): "Since you addressed my concerns thoroughly, I revised my score... The improvement ... is quite impressive."

Reviewer eWFW (Score: 4, **pending response**): *We provided exactly the real-world validation and metric justification requested (detailed below).*

With both reviewers fpEF and hbFR responding an *entire day* before the bug was publicly identified, we believe these interactions were unaffected and genuinely reflect the effectiveness of our rebuttal and the strength of our revised manuscript. That said, we believe our rebuttal arguments stands on its own merit, and we provide a high-level summary below.

----

### Key Technical Improvements in Rebuttal

Reviewers fpEF and hbFR's main initial reservations focused on computational cost and experimental scope. We resolved both issues during the rebuttal with significant improvements:

### A. Solved Scalability: $O\left(n^2m\right) \rightarrow O(nm)$ Complexity

 - We introduced a new A/B Grouping scheme that splits sources into two sets and cross-evaluates them. This structurally reduces complexity significantly to $O(nm)$ while strictly preserving the theoretical truthfulness guarantees. We demonstrated per-query costs are now considerably small  ( $<\$ 0.10$ ), and we re-ran all main experiments to prove this efficient scheme maintains the original performance gains. Both reviewers who initially raised the concerns agree that this resolves their concerns, with Reviewer hbFR commending this as *“quite impressive”*.

### B. Quadrupled Scale \& Real-World Verification

- Reviewer fpEF noted our initial evaluation (300 samples) was small.
We scaled our evaluation to the **entire** test sets of Natural Questions and ClashEval (over 2,400 items, a 4x increase). We also added a new experiment on an applicable subset of the QACC dataset (Liu et al., 2024) which uses real web pages. Even on this real-world data, TTS effectively downweighted unreliable sources, improving answer accuracy from ~46% (baselines) to 65.9%. Both reviewers agree that this resolves their concern, with reviewer hbFR noting that our expanded experiments *“make the empirical validation much stronger”*.

### C. Addressing Reviewer eWFW

Unfortunately, reviewer eWFW **didn’t have a chance to respond before comments were disabled**. We believe that we equally, if not more strongly, addressed their reservations:

- Evaluation Metrics: The reviewer questioned the accuracy of using gemini-2.5-flash as a judge. In our rebuttal, we demonstrated that for our specific binary verification tasks with a gold label, a SOTA deep-reasoning model is more consistent than the often-noisy crowdsourced human labels typically found in this domain. We supported this with examples and a manual check of outputs.

- Synthetic vs. Real Data: This concern was shared by Reviewer fpEF, who was persuaded by our rebuttal. We addressed this in two ways:
    - Current robustness: we added the requested real-world dataset (QACC), proving our method is effective against existing web noise,
    - Future validity: We justified that synthetic sources are necessary to model the *emerging* adversarial incentives arising from the increasing adoption of AI Overviews - incentives that are central to our paper's contribution but do not yet exist at scale on the web. Therefore, we believe our synthetic sources, generated by SOTA reasoning models, serve as the most relevant forward-looking proxy for this emerging threat landscape.

We believe these revisions comprehensively address the weaknesses identified by the reviewers. The paper now offers both a rigorous theoretical contribution and a feasible practical path for reliable AI search.

Please refer to our individual responses for details, and we thank the Area Chair again for your service.

---

### Meta-Review · Area_Chair_CXyk · 2026-01-08

**Summary:**

Key strengths all reviewers agreed:

1) The problem is relevant and timely; viewing it via game theoretic lens and designing TTS is nice.
2) Mathematical rigor
3) Promising results

Key weakness pointed out by most reviewers:

1) Cost: Multi-task peer prediction mechanism applied in LLM setting (with billion of queries per day) can be expensive
2) Empirical evaluation is not exhaustive and often on synthetic datasets.

----
To authors, I have few question out of curiosity based on my independent reading of the paper.
For AI summaries, most relevant RAG system is the Internet (search) itself. In this setting, why ranking (say page rank) itself is not a good indicator of truthfulness or accuracy of the website? In fact, this is precisely the role of ranking mechanisms in search. Given higher ranks to authoritative and correct sources of information.

Even in search ranking, one can tweak the information presented to maximize the ranking; so, how this problem is different from ranking setting? in particular, why LLM summaries should make a difference? We could think of top 10 blue links as the summaries of all internet, and viewed that way, does not ranking also have the same issue?  I see that they are not mathematically equivalent, but from a practical perspective they seem to be solving similar problems. Some discussions on these two fronts in the future revisions of the paper would be helpful to readers.

**Reviewer Concerns:**

The authors have put a lot of effort into addressing most of the reviewers' concerns. I think they did a good job of it.
Ofcourse, problem is too broad and difficult, so certainly there is plenty of room for improvement.
Most likely, the proposed mechanism is a bit impractical (for example we had similar issues in search, and I do not think search uses truthful mechanisms)

**Reviewer Scores:**

Most reviewers would have bumped their scores in the vicinity of 5-6. While one can debate about the assumptions, strengths and drawbacks, all reviewers would agree that 1) problem is relevant and is practical 2) community should study it as rigorously as possible, although solutions may not always make it real systems.

---

### Decision · Program_Chairs · 2026-01-26

Accept (Poster)